# Insight into the Association between Slitrk Protein and Neurodevelopmental and Neuropsychiatric Conditions

**DOI:** 10.3390/biom14091060

**Published:** 2024-08-26

**Authors:** Nidhi Puranik, Minseok Song

**Affiliations:** Department of Life Sciences, Yeungnam University, Gyeongsan 38541, Republic of Korea; nidhipuranik30@gmail.com

**Keywords:** Slitrk protein, leucine-rich repeats, neurodevelopment disorder, neuropsychiatric disorder, synaptogenesis

## Abstract

Slitrk proteins belong the leucine-rich repeat transmembrane family and share structural similarities with the Slits and tropomyosin receptor kinase families, which regulate the development of the nervous system. Slitrks are highly expressed in the developing nervous system of vertebrates, modulating neurite outgrowth and enhancing synaptogenesis; however, the expression and function of Slitrk protein members differ. Slitrk protein variations have been associated with various sensory and neuropsychiatric conditions, including myopia, deafness, obsessive–compulsive disorder, autism spectrum disorders, schizophrenia, attention-deficit/hyperactivity disorder, glioma, and Tourette syndrome; however, the underlying mechanism remains unclear. Therefore, the Slitrk family members’ protein expression, roles in the signaling cascade, functions, and gene mutations need to be comprehensively studied to develop therapeutics against neurodegenerative diseases. This study presents complete and pertinent information demonstrating the relationship between Slitrk family proteins and neuropsychiatric illnesses. This review briefly discusses neurodevelopmental disorders, the leucine-rich repeat family, the Slitrk family, and the association of Slitrk with the neuropathology of representative disorders.

## 1. Introduction

The human brain is complicated. During brain development, diverse cell types proliferate, differentiate into diverse paths, migrate to the correct sites, and integrate into functional circuitry. The human brain comprises 85 billion neurons and is capable of complex processes, such as language, cognition, and emotion recognition, which are affected by the growth process [1].

Neurodevelopmental disorders (NDDs) are brain-associated disorders that affect normal brain development and function through extensive inherent and clinical variability [2]. NDDs are characterized by various early clinical signs, such as developmental delay, cognitive/social impairment, and seizures. NDDs affect more than 3% of children worldwide [2]. Common NDDs include intellectual disabilities (ID), communication disorders, epilepsy, and neurodevelopmental motor disorders [3,4]. Therefore, neuropsychological research should explore multiple genetic factors because of the extremely polygenic nature of all behavioral, cognitive, and brain features [5]. Numerous NDD-related genes have been identified, with synaptic genes being the most prevalent [6]. NDDs are characterized by mutations in several genes related to the control of brain development and synaptic function [7]. Autism spectrum disorder (ASD), obsessive–compulsive disorder (OCD), schizophrenia (SZ), Tourette’s syndrome (TS), ID, and attention deficit/hyperactivity disorder (ADHD) are the most common NDDs. ASD is a pediatric psychiatric illness and an NDD. The main symptoms include confined and repetitive behaviors, interests, and activities and decreased social interaction and communication [8]. OCD is a common clinical mental condition that has a prolonged course, affects 2% of people in their lifetime, and is challenging to treat [9,10]. Anxiety disorder (AD) is a type of neurosis characterized by worry and is one of the top 10 most debilitating diseases. Affective disorders, such as depression, are common and detrimental to physical and mental health. OCD, AD, and depression are interconnected, changing depending on and restrained by one another. For example, OCD typically does not occur on its own. Most patients with OCD also experience anxiety, few experience sadness, and few experience AD and depression. These three disorders are caused by changes in or damage to part of the brain and nervous system [11].

The complex process of brain development makes the brain susceptible to various developmental disorders. The etiological factors and stage of brain development may interact to cause developmental problems [12]. Most trophic molecules may be involved in determining the specificity and flexibility of synapses. Neurotrophic factors (NTFs) are hypothesized to control the creation of excitatory synapses but not inhibitory synapses, and their signaling pathways appear to be evolutionarily conserved across all species. NTFs are specific proteins that aid in synaptic development, maturation at pre- and postsynaptic locations, and plasticity. These findings support the hypothesis of neuronal survival [13]. The interaction of receptors with their ligands and downstream cell signaling plays a crucial role in brain development. The leucine-rich repeat (LRR) protein family functions in nervous system (NS) development and regulation. Mammalian LAR-type RPTPs interact with various postsynaptic ligands, including TrkC, SALMs, Slitrks, IL-1RAPs, and netrin-G ligand 3 [14].

Synaptic Cell Adhesion Molecules (CAMs) play critical roles across multiple stages of synaptogenesis, which includes the creation of synapses, their maturation, refinement, plasticity, and elimination. Synaptic CAMs facilitate connections between pre- and postsynaptic compartments and play crucial roles in transsynaptic recognition and signaling processes. These processes are indispensable for establishing, specifying, and modulating synaptic plasticity. The family of synaptic CSM is expanding and includes neurexins, neuroligins, Ig-domain proteins like SynCAMs, receptor phosphotyrosine kinases, phosphatases, and various LRR proteins. Among the newly recognized CAM proteins, the Slitrk proteins of the LRR family are notable members. The LRR proteins and their families will be briefly addressed in the following section. However, this review focused on the Slitrk family proteins and their roles in neurodevelopmental disorders.

## 2. LRR-Domain Protein Family

The LRR protein family is one of the most abundant protein families and has been conserved from primitive organisms to humans. The LRR domain is present in approximately 330 proteins in the human genome and was first identified in immune system proteins as a protein–protein interaction motif. LRRs comprise 20–30 amino acid residues, with a highly conserved and a variable segment. It contains a conserved 11–12 amino-acid residue, LxxLxLxLxxN/CxL, where x is any amino acid [15,16].

LRR proteins contain a unique curved horseshoe-shaped structure made of a beta-strand and alpha-helix joined by loops. The LRR domain is a highly competent and adaptable motif in protein–protein interactions because the convex surface of the LRR is composed of alpha-helices that influence the curvature of the LRR domain, and the concave surface is composed of a continuous beta-sheet that offers an efficient ligand-binding site. Extracellular LRR-domain proteins are well suited for controlling intercellular communication and cell-to-cell adhesion. Although ligand identification is being performed frequently, the ligands of these extracellular LRR proteins remain largely unknown. According to research on invertebrate systems, extracellular LRR proteins influence important stages of neuronal growth and neural circuit construction, including axon guidance, target cell recognition, and synapse formation. The importance of these proteins is further underscored by the discovery that several LRR proteins are associated with neuropsychiatric and neurodevelopmental disorders in humans [17].

Surface proteins, including LRR proteins, interact with important presynaptic and postsynaptic proteins and are important players in the organization of excitatory and inhibitory synapses. They are essential in synapse formation, differentiation, and synaptic plasticity, because these proteins are important for functional brain activity and the changes or mutations leading to NDDs [18]. LRR proteins control the steering of axons and dendrites to their target locations during the early stages of CNS development. They facilitate the selection of suitable target cells within an area and the establishment of synaptic connections in these cells. LRR proteins control the myelination of axons and stabilization of neuronal circuits in the developing NS. Overall, these proteins are associated with human neurological and psychiatric illnesses, emphasizing their crucial functions in the establishment of brain circuits [19]. LRRK2 inhibition is a promising therapeutic strategy that has received considerable interest in the last 10 years [20]. A basic summary of the various LRR protein subfamilies LRRTM, Slitrk, FLRT, tropomyosin receptor kinase (Trk), NGL, and SALMs associated with synapse formation or synaptogenesis is presented in Table 1.

## 3. Structural Organization of SLITRK Proteins

A group of transmembrane proteins structurally related to the Slit and Trk protein receptors, known as the Slitrk family, has an external LRR domain that is identical to that of Slit and a brief intracellular region devoid of tyrosine phosphorylation motifs. The six members of the Slitrk family (Slitrk1–6) are expressed primarily in neural tissues [29] and control excitatory and inhibitory synapse formation via transsynaptic adhesions with LAR receptor protein tyrosine phosphatases (PTPs) PTPRD and PTPRS. Slitrks play a specific role in the maturation of a particular population of neurons at different stages of development; however, Slitrk6 differs from the other members due to its limited expression in the central nervous system (CNS) and its functions [30]. Slitrk gene studies in mice have shown that the genes are mainly expressed in neural tissues and exhibit a neurite-modulating action in cultured neuronal cells.

The basic structure of the Slitrk1–6 transmembrane protein is represented in Figure 1, and the general characteristics of Slitrk proteins, including protein Uniport ID number, chromosomal gene location, amino acid count, molecular weight, posttranslational modification type, majorly expressed tissue, neurological function, and associated neurological diseases, are listed in Table 2.

Recent investigations have provided some indications of the mechanism underlying its neurite-modulating activity. First, neurite outgrowth is inhibited in the intracellular domain. Second, this domain contains several phosphorylatable tyrosine (Tyr) residues that may be involved in signaling with proteins that possess SH2 domains. The carboxy-terminal domains of Slitrk and TrK are similar, suggesting that Slitrk is involved in PLC-mediated signaling, leading to an increase in intracellular calcium concentration by hydrolyzing phosphatidyl inositol bisphosphate into inositol triphosphate and diacylglycerol. Therefore, Slitrk2 can use calcium as a secondary messenger. Additionally, the signaling pathways shared by Slitrk and neurotrophin receptor proteins may overlap. Therefore, it is important to consider their potential to alter each other’s functionality [31]. Slitrks are connected to basic neuronal processes, such as dendritic elaboration, neurite development, and neuronal survival. Humans and genetic mouse model studies have identified Slitrks as the potential genes involved in the emergence of neuropsychiatric diseases (NSDs), including OCD and SZ. Slitrk may play an important role in CNS development, as demonstrated by system-level techniques [30].

Protein alignment and percentage identity index analyzed using UniProt (https://www.uniprot.org/align) showed that the six Slitrk proteins in humans have less than 50% identity with each other; however, only Slitrk2 and 5 showed approximately 52% identity. Because these proteins are present in vertebrates, the percent identity index was checked for the Slitrk protein sequence of commonly used experimental vertebrate models, including mice, Danre (zebrafish), and humans, for NSDs (https://www.uniprot.org/align). There is protein alignment of Slitrk1, 2, 4, and 6 for all mentioned experimental models; however, the Slitrk3 and Slitrk5 protein sequence data for zebrafish are not available on UniProt. Therefore, we only compared the similarity indices for humans, rats, and mice for these two proteins. The human Slitrk1 protein had a maximum identity of 97.56%, 97.41%, and 71.57% with mice, rats, and zebrafish, respectively. The human Slitrk2 protein had 97.63% identity with mouse and rat. The human Slitrk3 protein had 96.21% and 96.42% identity with rat and mouse, respectively. The human Slitrk4 protein had 73.13%, 97.49%, and 97.25% identity with zebrafish, rat, and mouse, respectively. The human Slitrk5 protein had 97.07% and 96.96% identity with mouse and rat, respectively. Among the six Slitrk proteins, Slitrk6 had a lower percentage identity with zebrafish, rat, and mice (54.22%, 88.10%, and 89.17%, respectively). The percentage identity results showed that most of these proteins remained conserved during evolution and speciation. Thus, these animal models are suitable for studying the role of Slitrk in human NSDs.

To understand the functions of Slitrks and their interactions with other proteins, the protein–protein interactions of closely related proteins were projected using the online String tool. Slitrk1, Slitrk2, and Slitrk3 interacted closely with PTPRD, PTPRS, PTPRF, NLGN1, NTR3, and IL1RAL1. However, Slitrk3 also interacted closely with NTRK1, NRXN1, and NRXN2; Slitrk4 interacted closely with PTPRD, NTRK1, ADGRL3, and EPHA6; Slitrk5 interacted closely with PTPRD, PTPRS, NTRK1, NTRK2, and RAC3; and Slitrk6 interacted closely with PTPRD and BDNF (Figure 2). All these proteins are involved in CNS development. The general characteristics and their roles in neurons are summarized in Table 3.

## 4. Slitrk and Neuro-Disease

Slitrks influence the establishment of inhibitory synapses and neurite outgrowth, according to mouse loss-of-function studies, although the molecular processes underlying this remain poorly understood [32]. Slitrks present abundant postsynaptic density (PSD) in rat brains, and its overexpression stimulates the development of new synapses, but RNAi-mediated Slitrk suppression reduces the density of existing synapses. Interestingly, Slitrks are necessary for the development of excitatory and inhibitory synapses in an isoform-dependent manner. However, Yim et al. discovered that Slitrk3 only worked at inhibitory synapses, whereas the others, except Slitrk6, primarily acted at excitatory synapses [33] (Figure 3). Slitrk2 and Slitrk5 have opposite control over excitatory and inhibitory synapse formation in dopamine neurons, and loss of function of Slitrk2 or Slitrk5 leads to hyperactive behavior [34]. Missense and non-synonymous mutations in Slitrk family members are associated with NDDs and NSDs (Table 4). Kang et al. (2016) studied missense mutations in Slitrk1, 2, and 4 associated with NSDs, which notably decreased Slitrk trafficking and synapse formation. Many single-amino-acid substitutions in Slitrk1 and Slitrk4 impaired the glycosylation of the Slitrks expressed, leading to impaired Slitrk trafficking and abrogated Slitrk binding to PTP⸹, a presynaptic adhesion molecule, in vitro. However, a single-amino-acid substitution in Slitrk2 affected synapse formation activity without affecting its expression in co-culture assays. Overall, mutations or dysfunctions in the Slitrk gene led to the inactivation of different cellular mechanisms that may be the basis of Slitrk-associated NSDs in humans [35].

### 4.1. Slitrk1

Slitrk1 participates in the formation of neurons, and its aberrant functions may contribute to the pathophysiology of NSDs, such as TS, ADHD, OCD [45], and trichotillomania [46]. Slitrk1 is associated with TS, a potentially crippling developmental neuropsychiatric condition characterized by the onset of motor and verbal tics and comorbid OCD and ADHD. By interacting with 14-3-3 proteins, which are abundantly expressed in the NS, Slitrk1 increases the dendritic development of the neuron [29,47].

Slitrk1 plays a role in controlling synapse development, evidenced by its presence in PSD fractions of excitatory synapses in mouse hippocampi. Beaubien et al. also showed the presence of Slitrk1 in excitatory synapses and confirmed that Slitrk1 participates in the development of excitatory synapses in vitro. The extracellular portion of Slitrk molecules has LRR domains that facilitate the establishment of synapses by interacting with LAR-type receptors [48].

A genetic association between Slitrk1 and Gilles de la Tourette syndrome (GTS) was established by Miranda et al. (2009). They tested two previously studied rare variants and three Slitrk1 genes in 154 individuals with GTS; the association between the variants and polymorphisms in the Slitrk1 gene is susceptible to GTS [49].

Katayama et al. (2009) studied the role of Slitrk1 in behavioral abnormalities. To study behavioral and neurochemical phenotypes, they generated Slitrk1-KO mice and found increased norepinephrine and 3-methoxy-4-hydroxyphenylglycol levels in mice with elevated anxiety or mood-related behavior [50].

Ozomaro et al. (2013) characterized genetic variation in Slitrk1 in patients with OCD. To observe the rare genetic variation, they sequenced the Slitrk1 coding exons of patients with OCD and control samples and identified three genetic variants, N400I, L63L, and T418S. To analyze the role of these variants, they performed functional analysis by transfecting these Slitrk1 variants into primary rat neuron culture. The overexpressed wild-type Slitrk1 stimulated neurite outgrowth to a significantly greater extent than the N400I mutant and control vector [37].

In 2019, Melo-Felippe et al. analyzed the allele and genotype frequency of Slitrk1 with four other functions (SAPAP3, PBX1, LMX1A, and RYR3) associated with OCD. The case study revealed a statistically noteworthy preponderance of the A allele in male patients compared to healthy males. Overall, the genotype distribution showed significant differences between patients with OCD who expressed symptoms and those who did not [51].

An experimental study by Hatayama and Aruga (2023) showed that Slitrk1 controls the growth of noradrenergic fibers through cell-autonomous and -nonautonomous mechanisms and partially modifies Sema3a-mediated neurite regulation. Slitrk1-KO mice exhibit anxiety-like behaviors due to neonatal dysregulation of the noradrenergic system, which can be reduced by an alpha-2 noradrenergic receptor agonist. Overall, NSD-related neuroplasticity may be caused by momentarily increased noradrenergic signaling during the newborn stage. Therefore, Slitrk1 may be a possible candidate for a genetic connection between neonatal noradrenergic signaling and NSD pathogenesis [52].

### 4.2. Slitrk2

Site-directed mutagenesis analysis revealed the role of Slitrk2 in synaptic differentiation and the formation of a synaptogenic complex with RPTP [53]. El and Chehadeh et al. (2022) revealed that Slitrk2 polymorphisms linked to NDDs resulted in impaired cognition and excitatory synaptic activity in mice. In an experimental study, they generated Slitrk2-KO mice that exhibit impaired long-term memory and abnormal behavior. The functional Slitrk2 protein is obligatory in the hippocampal CA1 region to mediate spatial reference memory in mice. Slitrk2-KO mice show abnormalities in spatial reference memory. These data indicate that Slitrk2 is involved in the development of neurodevelopmental disorders [41].

Loomis et al. described the interaction between Slitrk2 and membrane-associated guanylate kinases (MAGUKs) of the PSD95 subfamily. Co-immunoprecipitation of the postnatal mouse brain showed that PSD93 and PSD95 were connected to Slitrk2 in vivo. Slitrk2 directly interacted with PSD95 through a noncanonical Src homology 3 (SH3) domain binding motif. Slitrk2 was strongly clustered with PSD95 in 293T cells and ablation of the PSD95 SH3 domain or the Slitrk2 SH3 domain binding motif decreased this clustering. These findings support PSD95 as Slitrk2’s first recognized intracellular binding partner. Further research should examine whether interactions between Slitrk and MAGUK regulate Slitrk localization to synaptic locations and help recruit additional intracellular signaling molecules involved in postsynaptic differentiation [54].

According to Han et al. (2019), among the Slitrk family members, Slitrk2 directly interacts with PSD95 and Shank3, which have two excitatory scaffolds containing PDZ domains. Along with PSD95 and Shank3, Slitrk2 also forms in vivo complexes with MAGUK family members. The PDZ domain binding motif of Slitrk2 is necessary for excitatory synapse formation, transmission, and spinal development in the CA1 hippocampal area. These PDZ domain-mediated interactions also showed that Slitrk2 supports excitatory synapse formation and transmission in vitro and in vivo [55]. Katayama KI and the group studied the physiological roles of Slitrk2 in mice by generating Slitrk2-KO mice. The absence of the Slitrk2 protein led to anomalous neural network activity, altered synaptic integrity, and increased synaptic plasticity in the Slitrk2-KO mice [56].

### 4.3. Slitrk3

Slitrk3 induces only inhibitory synapses through PTPRD interactions, whereas the other Slitrks interact with PTPRS and PTPRD to create excitatory and inhibitory synapses [57]. The loss of Slitrk3 in vivo reduces the number of functional inhibitory hippocampal synapses but not excitatory synapses, confirming its role in controlling the development of inhibitory synapses [18].

Slitrk3 functions as a postsynaptic adhesion molecule that specifically controls the formation of inhibitory synapses by trans-interacting with axonal Trk PTP⸹. Mice lacking Slitrk3 had fewer inhibitory synapses in hippocampal CA1 neurons, increasing their susceptibility to seizures and spontaneous epileptiform activity. Therefore, the Slitrk3-PTP⸹ is an inhibitory-specific transsynaptic organizing complex necessary for the establishment of functional GABAergic synapses [58].

Neuroligin 2 (NL2) and Slitrk3 are necessary for the maturation of neurons and interact with NL2 to control the establishment of GABAergic synapses. Importantly, NL2 and Slitrk3 collaborate to form synapses via nanomolar affinity interactions of their extracellular domains. Disruption of the activity of the hippocampal network increases the probability of seizures due to a selective disruption of the NL2–Slitrk3 connection [59].

Efthymiou et al. (2022) revealed that the Slitrk3 mutation in humans is associated with GABAergic synapse development and impaired peripheral and CNS development. To experimentally investigate the role of Slitrk3, they generated two different mutations in the Slitrk3 gene that help regulate GABAergic synapses in hippocampal neuronal cultures and observed that these two mutations abolished the ability of Slitrk3 to enhance GABAergic synapse development [60].

### 4.4. Slitrk4

Among these Slitrk proteins, Slitrk4 has been less studied and its mechanism of action in neuronal development and differentiation is not well understood. Marteyn et al. (2011) reported that changes in Slitrk2 and Slitrk4 gene expression alter neurite growth and synaptogenic activity, and Slitrk2 gene knockdown is directly responsible for functional abnormalities in synapse development in cell culture. These neuropathological pathways may have therapeutic implications for DM1 (Myotonic dystrophy type 1)-related functional alterations in neuromuscular connections [61].

Perla et al. (2022) used sequencing to identify a novel gene variant for ASD in the Lebanese population and found a single-nucleotide variant (missense mutation) in the Slitrk4 gene on chromosome X of a male inherited from his mother, showing the potential of the Slitrk4 gene as a probable ASD candidate gene [42].

### 4.5. Slitrk5

Slitrk5 protein, widely expressed in the CNS, controls and participates in a number of crucial stages of synaptogenesis, neuron differentiation, axon and dendritic growth, and the degenerative process of the CNS. Slitrk5 has intricate relationships with numerous CNS illnesses, including gliomas of the brain, ADHD, ASD, and Parkinson’s disease. Therefore, Slitrk5 may be a potential therapeutic target for NDDs [62].

Slitrk5 is involved in physiological and pathological CNS processes, including neurite outgrowth, dendritic branching, synaptogenesis, and signal transmission in neurons. Controlling Slitrk5 expression can reduce neural function damage and facilitate the restoration of neural function, making Slirtk5 a suitable therapeutic candidate; however, it takes years to complete this process because the precise etiology and signaling mechanism of Slitrk5 in NDDs are not fully understood [62].

Song et al. (2015) demonstrated that Slitrk5 trans-interacts with PTP⸹ under basal conditions, but switches to a cis-interaction with the TrkB receptor in the presence of BDNF, which mediates its post-endocytic recycling and results in functional re-sensitization of neurotrophic signaling (Figure 4). Furthermore, Slitrk5 directly interacted with TrkB receptors to control BDNF-dependent physiological responses. TrkB exhibited a decrease in the rate of ligand-dependent recycling and a different response to BDNF administration in the absence of Slitrk5. By attracting the Rab11 effector protein Rab11-FIP3, Slitrk5 optimally directd TrkB receptors to recycle endosomes that are Rab11-positive. Thus, Slitrk5 mediates TrkB’s BDNF-dependent trafficking and signaling of TrkB as a co-receptor [63].

Song et al. (2017) revealed that a mutation in a synaptogenesis-associated gene of Slitrk5 leads to OCD. They sequenced the protein-coding part of the Slitrk5 gene of 377 patients with OCD for the analysis of rare non-synonymous mutations, compared it with the Genomes database, and successfully identified four rare non-synonymous mutations in patient samples. The identified mutations of the Slitrk5 gene diminished synaptogenic activity, evidenced using in silico studies and in vitro functional synaptogenesis assays. Overall, the rare functional mutations in Slitrk5 contribute to the genetic risk of OCD in humans [64].

Shmelkov et al. (2010) showed that Slitrk5 deficiency impairs corticostriatal circuitry and leads to OCD-like behavior in mice. Slitrk5 KO mice exhibit overactivation of the orbitofrontal cortex, abnormalities in striatal anatomy and cell morphology, and alterations in glutamate receptor composition, all of which contribute to deficient corticostriatal neurotransmission [65].

Zhang et al. (2015) examined the Slitrk5 gene of 377 affected TS candidates to observe the association between the Slitrk5 gene and TS using a single-nucleotide polymorphism (SNP) assay; however, the study was unable to prove that TS and Slitrk5 are related. Despite the lack of evidence for a relationship with SNPs tagging the Slitrk5’s coding area, risk alleles may have a function in controlling Slitrk5 expression [66].

Slitrk5 expression is associated with epilepsy; Slitrk5 expression was higher in the temporal neocortex of patients with temporal lobe epilepsy than that in non-epileptic individuals. Slitrk5 expression increased 24 h after status epilepticus in the temporal neocortex and hippocampus of rats with pilocarpine-induced epilepsy. These findings suggested that Slitrk5 may be associated with epilepsy, which could serve as a starting point for further investigation of the underlying connection between Slitrk5 and epilepsy and the therapeutic targets of antiepileptic medications [62].

Meyer (2014) studied 16 highly expressed genes within the CA1 region with possible implications for memory physiology in the hippocampal region CA1. Slitrk5 is one of the 16 selected genes that operate in corticostriatal synapses and may be involved in hippocampal physiology [67].

### 4.6. Slitrk6

Unlike other Slitrks, which are mostly expressed in brain tissues, Slitrk6 has a distinctive expression pattern in various organs, including sensory (ear cyst, retina, tongue, CNS, and epidermis) and non-sensory (limb buds, maxillary process, pharyngeal arches, cochlea, lung, gastrointestinal tract, etc.) tissues. These organs contain only a small area where Slitrk6 is expressed. Its expression is restricted to a few parts of the CNS, including the dorsal thalamus, cerebellum, and medulla. Slitrk6 is expressed in the thalamic compartment and is strongly associated with prosomere 2, which expresses Gbx2 [68].

Liu et al. genotyped 399 TS nuclear family trios to analyze SNPs in the Slitrk6 gene. A TDT (transmission disequilibrium test) did not show statistically significant allele transfer for the three polymorphisms. HRR (haplotype-relative-risk) was also negatively correlated to HHRR (haplotype-based haplotype-relative-risk). The study was unable to pinpoint the possible importance of Slitrk6 in the pathogenesis of TS despite the results that indicate that these polymorphisms may not be associated with susceptibility to TS in the Chinese Han population. Additionally, these findings need to be verified in multiple populations with larger sample sizes [69].

According to Matsumoto et al. (2011), Slitrk6 is essential for the growth of inner ear neuronal circuits. Slitrk6-KO mice show a marked reduction in cochlear innervation. The innervation of the posterior crista was often lost, diminished, or occasionally misdirected in the vestibule. The spiral and vestibular ganglia experienced loss of neurons along with these abnormalities. Slitrk6-deficient mice’s cochlear sensory epithelia are less effective in encouraging ganglion neurons to sprout neurites. Notably, the expression of BDNF and Ntf3, which are necessary for the innervation and survival of sensory neurons, was mildly but significantly reduced in the Slitrk6-deficient inner ear [70]. Additionally, in the Slitrk6-KO cochlea, NTRK receptor expression, including that of its phosphorylated form, decreased. Therefore, Slitrk6 regulates the production of trophic factors, such as neurotrophins, from sensory epithelia to increase the innervation and survival of inner ear sensory neurons [71].

The auditory phenotype of individuals with recessive Slitrk6 loss is described in detail for the first time. Nine Amish individuals who were related to each other closely conducted audiological and vestibular tests in an endogamous Amish community in Pennsylvania. High myopia, cochlear dysfunction attributed to outer hair cell disease, and progressive auditory neuropathy are associated with homozygous Slitrk6 c.1240C>T nonsense mutations [72]. A summary of experimental studies on the Slitrk protein for CNS-associated diseases is presented in Table 5.

## 5. Slitrk Protein-Related Study in Humans and Other Experimental Models

Slitrks are widely expressed in the mammalian CNS. Therefore, to study the role of Slitrk proteins in the development of NDDs and NSDs, various researchers have used experimental animal models, commonly mice, rats, and zebrafish. The percentage similarity index among the Slitrk proteins of these species also revealed great similarities among them. Various experimental studies conducted on human samples, mice, and zebrafish are briefly discussed.

### 5.1. In Humans

Human Slitrk1–6 genes were described by Aruga et al. in 2003. Six genes were distributed in three clusters on 3q, 13q, and Xq. Most of the expression was found in the brain, although each Slitrk had a different expression profile. Therefore, all Slitrk genes are differentially expressed in brain cancers, including supratentorial primitive neuroectodermal tumors (PNETs), oligodendrogliomas, glioblastomas, and medulloblastomas. Slitrk3 is widely expressed; however, Slitrk6 expression is extremely selective. These Slitrk genes can serve as valuable molecular markers of brain tumor characteristics [68].

The greatest signals for Slitrk1–5 were found in the cerebral cortex of adults, whereas Slitrk6 was found in the lungs and liver; however, Slitrk mRNAs were found primarily in neural tissue. Slitrk1 is highly expressed in the frontal lobe of the cerebral cortex, whereas Slitrk3 is highly expressed in the occipital pole. Slitrk2 and Slitrk5 were strongly expressed despite their low expression levels in the spinal cord and medulla. Adult lungs also express Slitrk6; virtually no additional gene expression was detected. However, Slitrk3 was distinguished in cancers, evidenced by its greater expression in brain tumors compared to the transcript content in the entire brain. The abundance of Slitrk3 ESTs (expressed sequence tags) in brain tissue was consistent with this characteristic. In PNETs, glioblastomas, and astrocytomas, Slitrk4 expression was also elevated, although it was more selective for benign tumors than Slitrk3. In contrast, Slitrk5 expression was the most widespread among many types of brain cancer. Unlike the other genes, Slitrk6 expression was amplified in PNET and in one case of medulloblastoma. Assuming that Slitrk6 expression was highly localized in specific regions of the brain, similar to mouse Slitrk6, and that the concentration of the transcript in the entire brain RNA was low, its transcript content may have been more than 40 times as abundant [75].

### 5.2. In Mice

The extracellular portion of Slitrks shares a high degree of similarity with the repellent axon guidance molecule Slits, further supporting the role of Slitrk in axon outgrowth throughout the development process. Beaubien et al. (2009) used in situ hybridization to show that Slitrks exhibited unique patterns of expression in different parts of the NS, and these various expression patterns implied that the members of the Slitrk family can play various roles during NS development [76].

### 5.3. In Zebrafish

Zebrafish express Slitrks in key regions and at critical moments of neuronal morphogenesis and synaptogenesis in the developing CNS. Using in situ hybridization techniques, Round et al. (2013) observed the spatial and temporal distributions of Slitrk gene expression in the midbrain, hindbrain, and spinal cord. Slitrk3b was expressed more in the head region; however, Slitrk1, Slitrk4, and Slitrk5 were selectively expressed in the midbrain. Slitrk2 was expressed in the neuroepithelium, pineal gland, and hindbrain. At 48 hpf, Slitrk3a and Slitrk5a showed the highest expression in telencephalon and diencephalon. They discovered eight Slitrk orthologs in zebrafish in a preliminary investigation of the functional activities of Slitrks and discovered that seven of the eight orthologs were actively translated into the NS in the embryonic, larval, and adult stages [32].

### 5.4. In Pigs

To establish similarities in the genetic organization of Slitrk1 in humans and pigs, Larsen et al. cloned and expressed the pig Slitrk1 gene. The porcine Slitrk1 protein comprised 695 amino acids, with a 99% homology to human Slitrk1. Only pig brain tissue expressed the Slitrk1 gene. The study concluded that pigs can be used as a viable model to investigate TS due to the high similarity of the Slitrk1 protein [77].

## 6. Slitrk and Other Diseases

Slitrk proteins are associated with cancer and are expressed in leukemia and lymphoma cells, as well as hematopoietic stem cells. Different Slitrks are expressed by the myeloid and lymphoid fractions of leukemic cells; lymphoid leukemias primarily express Slitrk1 and Slitrk6, whereas myeloid leukemias primarily express Slitrk4 and Slitrk5. Slitrk3 was not expressed in any of the leukemic cell lines examined (62). Slitrk is associated with cancer, for instance, Slitrk3 in lung cancer [78]; Slitrk4 in chronic myelomonocytic leukemia [79], gastric cancer, and liver metastasis [80]; Slitrk5 in left displacement of the abomasum [81] and papillary thyroid carcinoma [82]; and Slitrk6 in hepatocellular carcinoma [83], urothelial carcinoma [84], and lung adenocarcinoma [85]. Slitrk5 may be involved in bone formation and skeletal development as a negative regulator of Hedgehog signaling [86].

## 7. Expert Comment and Future Prospects

Various studies and evidence reviewed in this article show the association of Slitrk with neurodevelopment and changes in its expression or mutation lead to the development of NDDs or NSDs. However, more studies are required to clarify the specific and common roles of each member of the Slitrk family in vivo, which will contribute to a better understanding of NSDs.

There are a few outstanding questions raised by [30], and some remain unaddressed. Additional questions now need to be tackled. (i) The String results revealed a few common binding partners of the Slitrks. Are there more specific and unique binding partners in each Slitrk protein? (ii) If they have a common intra- and extracellular binding partner, how are their mechanisms of action different? How do Slitrks play a structural role in the synapse? Do Slitrks act as receptors for an unidentified ligand and trigger signaling cascades with their intracellular domains? Are more Slitrk variants associated with the development of NDSs and NSDs? As mentioned above, Slitrk5 negatively regulates Hedgehog signaling during bone formation. Does Slitrk5 also regulate Hedgehog signaling, a major signaling cascade in NS development? The role of Slitrk4 is not widely observed in NS development and has not yet been widely investigated.

These questions could be addressed by (i) the development of Slitrk knockout and knockin experimental models for each variant; (ii) genotyping (single polymorphism) of NDS and NDD patient samples; (iii) downregulating or overexpressing Slitrks mRNA in specific neuron cells; (iv) in silico studies to understand the protein–protein interaction/the binding partner; or (v) expression profiling of each SLITRK member in the interaction of proteins that generates new datasets that provide more possibilities for interpretation of function.

## 8. Conclusions

Numerous processes are required for NS development, including synapse formation and refinement, axonal guidance, cellular migration, and cellular differentiation. The NS exhibits significant levels of LRR expression, and may control different stages of neuronal growth. Slitrk proteins are a large subfamily of transmembrane proteins that contain LRRs and have various intracellular domains, which may allow them to play various roles during neuronal formation. The role of Slitrk family members during NS development is still largely understood; however, these proteins were initially discovered in a screen for differentially expressed genes in mice with neural tube abnormalities. Strong efforts must be made to thoroughly investigate the functions of Slitrks during early NS development and their control in later stages.

## Figures and Tables

**Figure 1 biomolecules-14-01060-f001:**
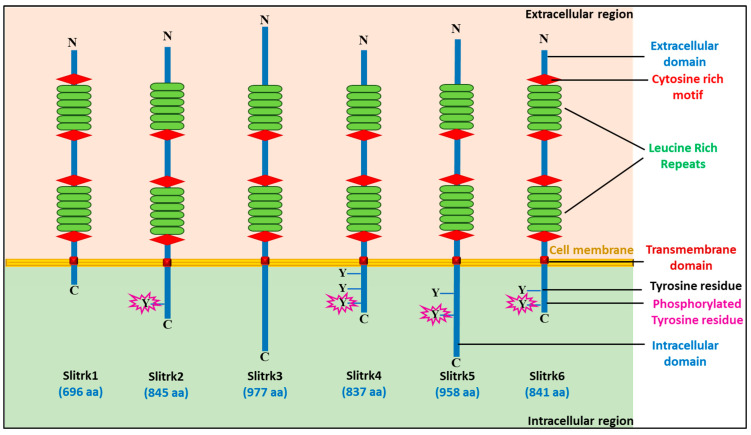
Basic schematic representation of the Slitrk1–6 transmembrane protein. Slitrk proteins have an extracellular domain containing LRR motifs that vary in number, flanked by cysteine-rich motifs, followed by a single transmembrane domain and an intracellular domain. The conserved tyrosine (Y) in the intracellular domain is phosphorylated, as shown in Slitrk2, Slitrk5, and Slitrk6.

**Figure 2 biomolecules-14-01060-f002:**
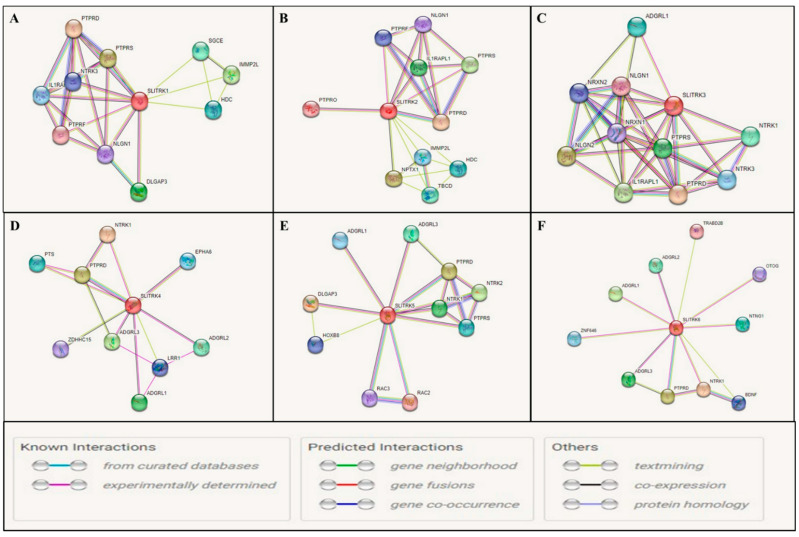
Protein–protein interaction analysis for Slitrk1–6. (**A**) Slitrk1, (**B**) Slitrk2, (**C**) Slitrk3, (**D**) Slitrk4, (**E**) Slitrk5, and (**F**) Slitrk6. The intricate organization and regulation of synaptic connections involve complex and coordinated molecular and cellular processes. Diverse protein–protein interactions and their functional impacts at synapses. To explore the functions of Slitrks and their interactions with other proteins, we utilized the online String tool to predict protein–protein interactions among closely related proteins. Slitrk1, Slitrk2, and Slitrk3 were closely associated with PTPRD, PTPRS, PTPRF, NLGN1, NTR3, and IL1RAL1. Notably, Slitrk3 also exhibited close interactions with NTRK1, NRXN1, and NRXN2; Slitrk4 showed close interactions with PTPRD, NTRK1, ADGRL3, and EPHA6; Slitrk5 demonstrated close interactions with PTPRD, PTPRS, NTRK1, NTRK2, and RAC3; and Slitrk6 interacted closely with PTPRD and BDNF.

**Figure 3 biomolecules-14-01060-f003:**
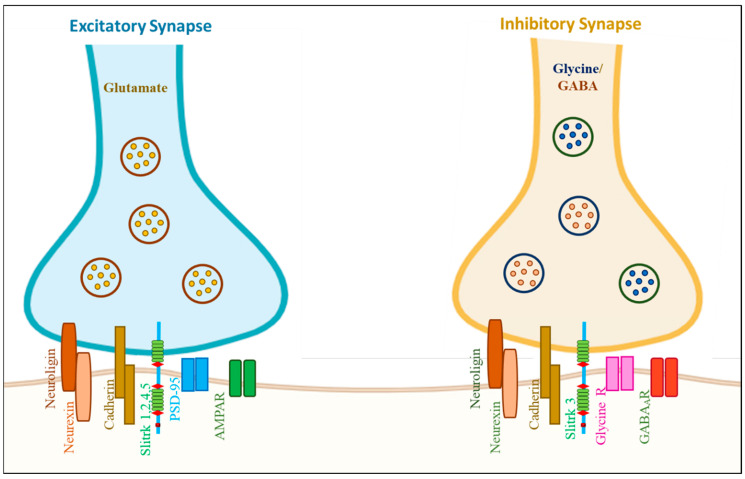
Graphical representation of the molecular organization of excitatory and inhibitory synapses. The regulation of neural circuit functions and behaviors is dictated by numerous synaptic organizers that maintain the equilibrium between excitatory and inhibitory synaptic inputs. Among Slitrk family proteins, certain Slitrk isoforms, Slitrk1, 2, 4, and 5, were found to selectively function at excitatory synapses in diverse functional assays, whereas Slitrk3 specifically operated at inhibitory synapses. As shown in the figure, Slitrk1, 2, 4, and 5 present excitatory synapses; however, Slitrk 3 is an inhibitory synapse as an adhesion molecule (adopted from [36] and created by Biorendor.com).

**Figure 4 biomolecules-14-01060-f004:**
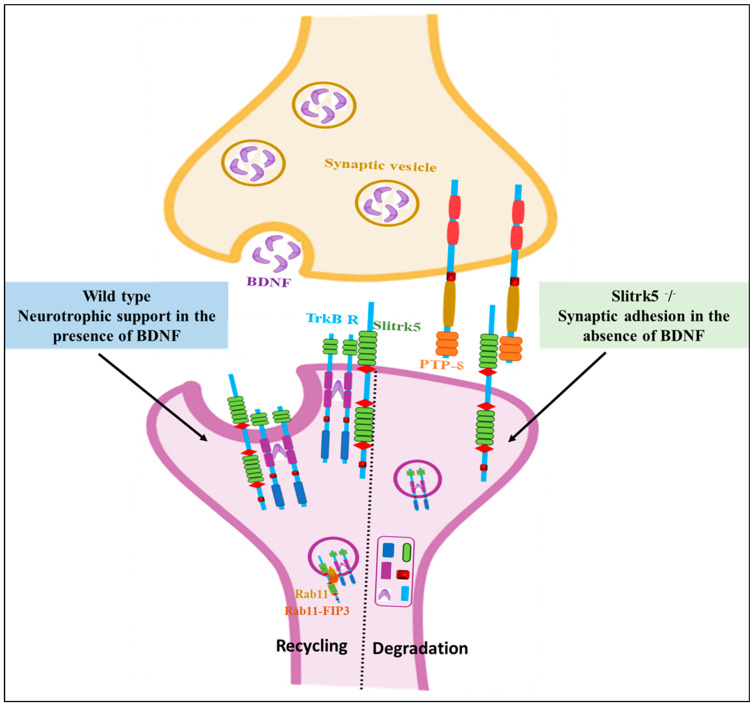
BDNF signaling and Slitrk5 (adopted by [49] and created by Biorendor.com). Slitrk5, a cell-surface protein with LRR motifs, acts as a co-receptor that enhances TrkB signaling. TrkB and Slitrk5 physically interact by forming a complex involving their LRR domains. TrkB and PTPδ compete for binding to the initial LRR domain of Slitrk5. Slitrk5 influences BDNF-mediated biological reactions by directly controlling the recycling of the TrkB receptor through the recruitment of Rab11-FIP3.

**Table 1 biomolecules-14-01060-t001:** Summary of the LRR protein family.

LRR Family Name	Family Members	LRR Domain	Expressed in Human Tissue/Organ	Function	Commonly Associated Disease	References
LRRTM Family	4 (LRRTM1–4)	Ten LRR motifs	In entire brain	Formation of excitatory synapses; stimulate excitatory synaptogenesis	SZ, ASD, and AD	[21]
SLITRK Family	6 (SLITRK1–6)	Six LRR motifs	Brain and other tissues too	Neuritogenesis	TS, OCD, and ADHD	[21]
FLRT (Fibronectin leucine-rich repeat) Family	3 (FLRT1–3)	Ten LRR motifs	Many tissues, including the brain	Axon guidance and cell migration	ADHD	[22]
Trk (Tropomyosin receptor kinase) Family	3 (TrkA-C)	Three LRRs	Widely expressed throughout the NS	Axonal and dendritic growth and remodeling, synapse maturation and plasticity	AD and PD	[23,24]
NGL (Netrin-G ligand; LRRC4) family	3 (NGL1–3)	Nine LRRs flanked by cysteine-rich domain	Postsynaptic membrane	Regulate the formation of excitatory synapses and; the development of axons, dendrites, and synapses	PD	[25,26]
SALMs (Synaptic adhesion-like molecules) Family	5 (SALM1–5)	Six LRRs motif	Majorly expressed in the brain and various brain regions	Regulate synapse formation	ASD	[27,28]

**Table 2 biomolecules-14-01060-t002:** General characteristics of Slitrk proteins (data retrieved from UniProt and Genecards).

Protein Name (Uniport ID No.)	Gene Location	No of Amino Acid and MW	Protein 3D Structure	Type of PTM (Amino Acid Number)	Majorly Expressed in	Neurological Function	Associated Neurological Disease
Slitrk1 (Q96PX8)	Chromosome 13 (13q31.1)	696 77.7 kDa	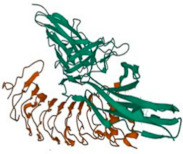	Phosphoserine (695)	Restricted to brain	Enhances the differentiation of excitatory synapses and is involved in synaptogenesis; Increases the development of neuronal dendrites	Neuropsychiatric disorder (OCD)
Slitrk2 (Q9H156)	Chromosome X (Xq27.3)	845 95.4 kDa	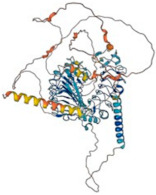	N-linked-Glycosylation (84 and 421) Disulfide bond (220–243 and 222–263) Phosphotyrosine (756)	Brain and spleen	It contributes to excitatory synapse differentiation during synaptogenesis and inhibits neurite proliferation	Found in a patient with SZ (a serious mental disorder)
Slitrk3 (O94933)	Chromosome 3 (3q26.1)	977 108.9 kDa	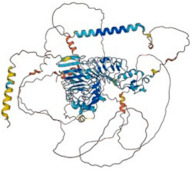	N-linked-Glycosylation (68 and 596)	Brain and esophagus	Suppresses neurite outgrowth	NA
Slitrk4 (Q8IW52)	Chromosome X (Xq27.3)	837 94.3 kDa	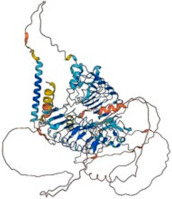	N-linked-Glycosylation (81 and 325) Phosphoserine (663 and 741) Phosphotyrosine (742, 769 and 811)	Adrenal gland and brain	It contributes to synapse differentiation and synaptogenesis, and it inhibits neurite outgrowth	Found in a patient with SZ
Slitrk5 (O94991)	Chromosome 13 (13q31.2)	958 107.4 kDa	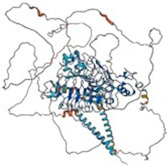	N-linked-Glycosylation (103 and 644) Phosphoserine (846) Phosphotyrosine (932 and 945)	Brain and thyroid gland	Suppresses neurite outgrowth	OCD, ADHD, glioma, ASDs, PD
Slitrk6 (Q9H5Y7)	Chromosome 13 (13q31.1)	841 95.1 kDa	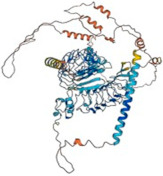	Phosphoserine (652 and 728) Phosphotyrosine (820 and 833)	Brain, urinary bladder, and salivary gland	Regulator of neurite outgrowth	Hearing loss; associated with high myopia

**Table 3 biomolecules-14-01060-t003:** String protein–protein interaction (data retrieved from String).

Protein	Interaction with Slitrk Members	Other Interactive Proteins	Function in Neuron
PTPRD	Slitrk 1–6	PPFIA1, PPFIA2, PPFIA3 IL1RAP and IL1RAPL1	Induce both pre- and postsynaptic differentiation of neurons; by mediating interaction with transsynaptically
PTPRS	Slitrk 1,2,3 and 5	NTRK3, NTRK1, PPFIA1, PPFIA2 and PPFIA3	Inhibition of neurite and axonal outgrowth; essential for normal brain development
DLGAP3	Slitrk 1 and 5	DLG4/PSD-95	Molecular organization of synapses and neuronal cell signaling
PTPRF	Slitrk 1 and 2	GRIP1, PPFIA1, PPFIA2 and PPFIA3	Involved in presynaptic differentiation
IL1RAPL1	Slitrk 1,2 and 3	NCS1, PTPRD	Neurite outgrowth
NTRK3	Slitrk 1 and 3	PTPRS, QSTM1 and KIDINS220	Involved in the NS development
NLGN1	Slitrk 1,2 and 3	NRXN1, NRXN2 and NRXN3	Synapse function and signal transmission
NRXN2	Slitrk3	NLGN1, NLGN2 and NLGN3	Involved in cell recognition and cell adhesion

**Table 4 biomolecules-14-01060-t004:** Studies based on a mutation in Slitrk family members that are associated with NDDs/NSDs.

Slitrk Member	Approach	Mutation Type and Amino Acid Substitution	Associated Disease	Summary	References
Slitrk1 Slitrk4 Slitrk2	In silico study	Missense N400I, T418S Missense V206I, I578V Missense V89M	NSDs- OCD and SZ	Invitro study—impaired glycosylation of Slitrks, impaired trafficking in neurons; eliminated Slitrk binding to PTP⸹ leads to inhibition of their function on synapse density; impaired synapse formation in cultured neurons	[35]
Slitrk1	Genome sequencing	Synonymous L63L Missense N400I and T418S	OCD	N400I mutation abolishes induced neurite outgrowth	[37]
Slitrk5	Genome sequencing	Non-synonymous mutations N99K, Q118H, E609K, G722∆, A851V, and P891L	OCD	All mutations cause impaired synaptogenic activity in OCD	[38]
Slitrk1	Genome sequencing	Deletion (frameshift mutation) var321	TS	-	[39,40]
Slitrk2	Genome sequencing	Missense mutation L74S, V201I, and E210K, P374R, R426C, R484Q, V511M, and E555D Nonsense mutation E461	ID	In vitro study, variations inhibit the functional activity of Slitrk2 and reduce its binding to TrkB in neurons	[41]
Slitrk4	Whole-exome sequencing	Missense mutation c.1860A>C p.Leu620Phe	ASD	Potential ASD candidate genes	[42]
Slitrk1	Sequence analysis	Missense mutation 383g>a	TS	Slitrk1 associated with TS pathophysiology	[43]
Slitrk6	Microarray and pyrosequencing	c.1232C>G, p.Thr411Arg	TS	Alter the opioid pathway	[44]

**Table 5 biomolecules-14-01060-t005:** Summary of experimental studies showing the association between Slitrk protein and CNS-associated diseases.

Slitrk	Disease	Model	Target Gene	Method	Outcome	Reference
Slitrk1	TS	Human clinical sample	Slitrk1 var321	Var321 genotyping	Slitrk1 var321 might be associated with TS; however, it is not associated with either TS or OCD within this sample	[40]
Slitrk1	TS	TS patient	Slitrk1 mRNA and hsa-miR-189	DNA mapping	Frameshift mutation (deletion) resulted in a truncated protein that led to TS	[73]
Slitrk1	OCD	Human	322 OCD probands for Slitrk1 var321 and varCDfs	Genotyping assays	var321, varCDfs, and SLC6A4 G56A have shown no direct connection with OCD	[74]
Slitrk1	OCD	OCD patient	Slitrk1 gene mutation	Genetic screening	A synonymous L63L change and a missense mutation T418S were identified in OCD individual	[37]
Slitrk2	ADHD	Slitrk2-KO mice	Slitrk2 null mutation	Electrophysiological analysis and behavioral study	Shows hyperactivity with altered vestibular function and serotonergic dysregulation	[56]
Slitrk2 and Slitrk5	ADHD	Lmx1a/b mice	Lmx1a/b target genes	Electrophysiological analysis and behavioral study	Mutation or Slitrk2 KO causes hyperactivity behavior	[34]
Slitrk4	ASD	23 male Lebanese ASD subjects	Single nucleotide variations in the Slitrk4 gene	Whole-Exome Sequencing	p.Leu620Phe variation in Slitrk4 is as potential ASD candidate genes	[42]
Slitrk4	DM1	Neural cells and DM1 brain biopsies	Slitrk4 gene	RT-PCR (gene expression)	Downregulation of Slitrk4 leads to change in neurite outgrowth and synaptogenesis	[61]
Slitrk5	TS	377 affected children	Slitrk5 gene	SNP	No direct correlation between TS and Slitrk5	[66]
Slitrk5	TLE and	TLE patients and epilepsy rat model	Slitrk5 Gene and protein	Gene expression	Change in Slitrk5 expression leads to epilepsy	[62]
Slitrk5	OCD	OCD mouse	Slitrk5 gene	Behavioral study	This shows up as heightened anxiety-like behaviors and obsessive self-care	[65]
Slitrk5	OCD	Slitrk5-KO mice	Slitrk5 gene	Genomes database-based study and functional study in mice	Mutations in the coding gene of Slitrk5 associated with OCD	[38]
Slitrk6	TS	Human samples	Slitrk6 gene	SNP	No statistically significant allele transfer	[69]
Slitrk6	Ear development	Slitrk6-KO mice	mRNA profiling and Histological examination	Associated with auditory and vestibular sensory organ functions in experimental mice	Slitrk6 regulates the production of tropic factors, such as neurotrophins responsible for the survival of inner ear sensory neurons	[71]
Slitrk6	Auditory-Vestibular functioning	Slitrk6-KO mice	Auditory function and vertical vestibular function test	Systematic behavioral and auditory-vestibular functioning	Head-dip behavior increased and no other clear abnormalities were noted	[70]
Slitrk6	Myopia	9 patients with myopia and Slitrk6−/−) mice	Slitrk6	SNP, genotyping, and genetic mapping and Auditory and Vestibular Testing	Slitrk6 homozygote c.1240C>T High myopia, cochlear dysfunction, and progressive auditory neuropathy	[72]

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
