# Peer review of "Insight into the Association between Slitrk Protein and Neurodevelopmental and Neuropsychiatric Conditions"

_biomolecules, 2024, doi:10.3390/biom14091060_

Round 1

Reviewer 1 Report

Comments and Suggestions for Authors

The manuscript provides a comprehensive and detailed review of the role of Slitrk proteins in neurodevelopmental and neuropsychiatric disorders. It effectively highlights the significance of these proteins in nervous system development and their association with various conditions such as ASD, OCD, schizophrenia, Tourette syndrome, intellectual disabilities, and ADHD. This review article is well-structured, covering the structural organization of Slitrk proteins, their functions, interactions, and the impact of their mutations on neurodevelopmental disorders. The manuscript also mentioned the role of Slitrk proteins in cancer biology, which adds an interesting dimension to their significance.

My concerns for this manuscript are listed as below

1.      Ensure consistency in terminology and correct any typographical errors.

Line 29: processedàprocesses

Line 72: LLRàLRR

Line 188: theyàthe

Table 2: UniportàUniprot

Line 146: SlickàSlitrk

Table 4: SDàSZ (?) (Schizophrenia? )

Line 234: “ was performed” should be deleted

2.    Ensure that all abbreviations are defined the first time they appear in the text.

Line 312: DM1àmyotonia dystrophy type 1

Line 381: TDTàtransmission disequilibrium test

Line 382: HRRàhaplotype relative risk

Line 382: HHRRàhaplotype-based haplotype relative risk

Line 431: ESTsàexpressed sequence tags

3.      About the study of Ozomaro et al, (2013) (mentioned in line 231), they transfected these Slitrk1 variants into “primary rat neuronal cultures”, not “into primary rat”, it should be clarified.

4.  While the manuscript is very comprehensive, some sections could be more concise. Reducing redundancy and focusing on the most critical findings would enhance readability.

5. More detailed mechanistic insights into how these Slitrk proteins influence neurodevelopmental and neuropsychiatric disorders would be beneficial, but it's just optional. 

Comments on the Quality of English Language

The quality of English in this manuscript is fluent and readable, but several typos and grammatical errors that I have highlighted in the suggestions for the authors should be corrected.

Author Response

Comments and Suggestions for Authors

The manuscript provides a comprehensive and detailed review of the role of Slitrk proteins in neurodevelopmental and neuropsychiatric disorders. It effectively highlights the significance of these proteins in nervous system development and their association with various conditions such as ASD, OCD, schizophrenia, Tourette syndrome, intellectual disabilities, and ADHD. This review article is well-structured, covering the structural organization of Slitrk proteins, their functions, interactions, and the impact of their mutations on neurodevelopmental disorders. The manuscript also mentioned the role of Slitrk proteins in cancer biology, which adds an interesting dimension to their significance.

Dear reviewer:

We appreciate the editorial staff and reviewers for your time and effort in reviewing the manuscript “Insight into the Association Between Slitrk Protein and Neurodevelopmental and Neuropsychiatric Conditions”, and give us an opportunity to resubmit the revised version. The comments provided by reviewers were valuable and helped us refine our paper. We have responded to the reviewers' questions point to point and highlighted the changes in yellow in the revised manuscript.

My concerns for this manuscript are listed as below

  1. Ensure consistency in terminology and correct any typographical errors.

Response: We thank the reviewer for pointing this out.

Line 29: processedàprocesses

Line 72: LLRàLRR

Line 188: theyàthe

Table 2: UniportàUniprot

Line 146: SlickàSlitrk

Table 4: SDàSZ (?) (Schizophrenia? )

Line 234: “ was performed” should be deleted

Response: Thank you point out this major issue. All the typo errors are corrected. The English language company “editage” proofread the paper to minimize the language issues.

  1. Ensure that all abbreviations are defined the first time they appear in the text.

Line 312: DM1àmyotonia dystrophy type 1

Line 381: TDTàtransmission disequilibrium test

Line 382: HRRàhaplotype relative risk

Line 382: HHRRàhaplotype-based haplotype relative risk

Line 431: ESTsàexpressed sequence tags

Response: We thank the reviewer for highlighting this point. All the full form is added in the revised MS and also a list of abbreviations is mentioned in the revised MS.

  1. About the study of Ozomaro et al, (2013) (mentioned in line 231), they transfected these Slitrk1 variants into “primary rat neuronal cultures”, not “into primary rat”, it should be clarified.

Response: Thanks for the reviewer’s question. It is a primary rat neuron that is cultured. The sentence is clarified in the revised MS.

  1. While the manuscript is very comprehensive, some sections could be more concise. Reducing redundancy and focusing on the most critical findings would enhance readability.

Response: We agree with the reviewer's suggestions. We tried to concise the MS accordingly.

  1. More detailed mechanistic insights into how these Slitrk proteins influence neurodevelopmental and neuropsychiatric disorders would be beneficial, but it's just optional. 

Response: Thank you for your valuable suggestions. However, the mechanistic details of Slitrk proteins are not yet fully understood. We have compiled all available relevant studies that explore the potential mechanisms by which Slitrk proteins may be involved in neurodevelopmental and neuropsychiatric disorders.

While we recognize the current limitations in detailed molecular analyses in the field, we have emphasized in the Expert Comment and Future Prospects section the potential directions for future research, emphasizing the importance of uncovering these mechanisms for a deeper understanding of Slitrk protein biology.

 Comments on the Quality of English Language

The quality of English in this manuscript is fluent and readable, but several typos and grammatical errors that I have highlighted in the suggestions for the authors should be corrected.

Response: We agree with the reviewer's suggestions. A complete proofreading is done before submission of the revised MS.

We hope the above responses have clarified our approach and addressed all your concerns. We are grateful for your constructive feedback and appreciate your consideration of our manuscript.

Reviewer 2 Report

Comments and Suggestions for Authors

This review article provides a comprehensive overview of the current knowledge on the Slitrk family of proteins involved in synapse formation. The number of references cited is sufficient, and the authors have approached the topic comprehensively. The article delivers a solid review of the current knowledge about Slitrk proteins and their role in synaptogenesis, albeit with some gaps in the detailed analysis of molecular mechanisms, interactions with other proteins, and potential therapies. These additional details could significantly enhance the scientific value of the review and its usefulness for readers interested in this research area.

 Critical comments

1. While the manuscript discusses many aspects of the structure and function of the Slitrk protein family, with particular emphasis on the association of their dysfunction with various disorders, it lacks a detailed analysis of the molecular mechanisms underlying these associations. The authors could provide a more thorough discussion of how mutations in Slitrk-coding genes affect the structure and function of synapses and which specific signaling pathways are involved in these processes.

2. Additionally, the article could include more information on the interactions of Slitrk proteins with other synaptic proteins. Although a few key interaction partners are mentioned, there is a lack of a comprehensive review of all known interactions and their functional consequences.

3. Slitrk proteins belong to Synaptic Cell Adhesion Molecules (CAMs), which determine their function and explain their ubiquitous presence in the body. Please include this information in the Introduction and mention other proteins that play a significant role in achieving mechanistic cell-cell recognition and initiating synapse formation.

4. Given the large number of abbreviations used, including a list of abbreviations would significantly facilitate reading the paper. Please provide a list of abbreviations.

5. Figures 2, 3, and 4 should have a more detailed description. Including a legend for these figures would be helpful.

Author Response

Comments and Suggestions for Authors

This review article provides a comprehensive overview of the current knowledge on the Slitrk family of proteins involved in synapse formation. The number of references cited is sufficient, and the authors have approached the topic comprehensively. The article delivers a solid review of the current knowledge about Slitrk proteins and their role in synaptogenesis, albeit with some gaps in the detailed analysis of molecular mechanisms, interactions with other proteins, and potential therapies. These additional details could significantly enhance the scientific value of the review and its usefulness for readers interested in this research area.

Dear reviewer:

We appreciate the editorial staff and reviewers for your time and effort in reviewing the manuscript “Insight into the Association Between Slitrk Protein and Neurodevelopmental and Neuropsychiatric Conditions”, and give us an opportunity to resubmit the revised version. The comments provided by reviewers were valuable and helped us refine our paper. We have responded to the reviewers' questions point to point and highlighted the changes in yellow in the revised manuscript.

Critical comments

  1. While the manuscript discusses many aspects of the structure and function of the Slitrk protein family, with particular emphasis on the association of their dysfunction with various disorders, it lacks a detailed analysis of the molecular mechanisms underlying these associations. The authors could provide a more thorough discussion of how mutations in Slitrk-coding genes affect the structure and function of synapses and which specific signaling pathways are involved in these processes.

Response: Thank you for your thoughtful evaluation of our manuscript. We agree with your point that understanding these molecular mechanisms is crucial for comprehensively elucidating the role of Slitrk proteins in synaptic function and dysfunction. However, it is important to note that the current state of research on Slitrk proteins has limitations in terms of detailed studies on specific molecular pathways affected by mutations in Slitrk-coding genes. The manuscript already covers a comprehensive review of the existing literature on Slitrk proteins, including their structural and functional aspects, as well as their implications in disorders. We have highlighted the BDNF pathway as one of the known mechanisms, which is discussed in detail within the manuscript. In our laboratory, we are actively investigating Slitrk4 and Slitrk5, aiming to uncover more precise molecular mechanisms associated with these proteins. We are optimistic that our ongoing research will contribute significant insights into how mutations in these genes impact synaptic structure and function, and the specific signaling pathways involved.

While we recognize the current limitations in detailed molecular analyses in the field, we have emphasized in the Discussion section the potential directions for future research, emphasizing the importance of uncovering these mechanisms for a deeper understanding of Slitrk protein biology.

  1. Additionally, the article could include more information on the interactions of Slitrk proteins with other synaptic proteins. Although a few key interaction partners are mentioned, there is a lack of a comprehensive review of all known interactions and their functional consequences.

Response: Again we show our gratitude to the reviewers for suggesting important points.

To understand the functions of Slitrks and their interactions with other proteins, the protein-protein interactions of closely related proteins were projected using the online String tool. Slitrk1, Slitrk2, and Slitrk3 interacted closely with PTPRD, PTPRS, PTPRF, NLGN1, NTR3, and IL1RAL1. However, Slitrk3 also interacted closely with NTRK1, NRXN1, and NRXN2; Slitrk4 interacted closely with PTPRD, NTRK1, ADGRL3, and EPHA6; Slitrk5 interacted closely with PTPRD, PTPRS, NTRK1, NTRK2, and RAC3; and Slitrk6 interacted closely with PTPRD and BDNF. All these proteins are involved in NS development. The general characteristics and their roles in neurons are summarized in Table 3.

However, for all these proteins except a few (PTPRD, PTPRS, BDNF) no published articles are available that have been showing the molecular and cellular mechanisms of their interaction in synapse development and regulation.

  1. Slitrk proteins belong to Synaptic Cell Adhesion Molecules (CAMs), which determine their function and explain their ubiquitous presence in the body. Please include this information in the Introduction and mention other proteins that play a significant role in achieving mechanistic cell-cell recognition and initiating synapse formation.

Response: Thank you for your constructive feedback on our manuscript. We appreciate your suggestion to include information about Slitrk proteins as Synaptic Cell Adhesion Molecules (CAMs) in the Introduction, along with mentioning other proteins that contribute to mechanistic cell-cell recognition and synapse formation.

Synaptic Cell Adhesion Molecules (CAMs) play critical roles across multiple stages of synaptogenesis, which includes the creation of synapses, their maturation, refinement, plasticity, and elimination. Synaptic CAMs facilitate connections between pre- and postsynaptic compartments and play crucial roles in trans-synaptic recognition and signaling processes. These processes are indispensable for establishing, specifying, and modulating synaptic plasticity. The family of synaptic CSM is expanding and includes neurexins, neuroligins, Ig-domain proteins like SynCAMs, receptor phosphotyrosine kinases, phosphatases, and various LRR proteins. Among the newly recognized CAM proteins, the Slitrk proteins of the LRR family are notable members.

  1. Given the large number of abbreviations used, including a list of abbreviations would significantly facilitate reading the paper. Please provide a list of abbreviations.

Response: We agree with the reviewer's suggestion. A list of all abbreviated forms is mentioned in the revised MS.

Abbreviations- NDDs- Neurodevelopmental disorders; NSDs- Neuropsychiatric diseases; NS-Nervous system; AD- Anxiety disorder; ASD- Autism spectrum disorders; OCD- Obsessive-compulsive disorder; SZ- Schizophrenia, TS- Tourette syndrome; ADHD- Attention-deficit/hyperactivity disorder; ID- Intellectual disability, TLE- Temporal lobe epilepsy; SE- Status epilepticus; CAMs- Cell Adhesion Molecules; GTS- Gilles de la Tourette Syndrome; PTPRD- Receptor-type tyrosine-protein phosphatase delta; PTPRS-Receptor-type tyrosine-protein phosphatase S; DLGAP-Disks large-associated protein 3; PTPRF-Receptor-type tyrosine-protein phosphatase F; IL1RAPL1-Interleukin-1 receptor accessory protein-like 1; NTRK1-NT-1 growth factor receptor; NTRK2-NT-2 growth factor receptor; NTRK3-NT-3 growth factor receptor; NLGN1-Neuroligin; NRXN2-Neurexin-2; NTFs- Neurotrophic factors; LRR- Leucine-rich repeat; Trk- Tropomyosin receptor kinase; CNS- Central nervous system; ADGRL3- Adhesion G Protein-Coupled Receptor L3;  EPHA6- EPH Receptor A6; RAC3- Rac Family Small GTPase 3; PSD- Postsynaptic densities; BDNF- Brain-derived neurotrophic factor; MAGUKs- Membrane-associated guanylate kinases; KO- Knockout; DM1 Myotonic dystrophy type 1; TDT- transmission disequilibrium test; HRR- Haplotype-relative-risk; HHRR- Haplotype-based haplotype-relative-risk.

  1. Figures 2, 3, and 4 should have a more detailed description. Including a legend for these figures would be helpful.

Response: Thank you for your constructive feedback. We agree with the reviewer's suggestion. In the revised MS, we have added a description for figures 2, 3, and 4.

Figure 2. Protein-protein interaction analysis for Slitrk1–6. A) Slitrk1, B) Slitrk2, C) Slitrk3, D) Slitrk4, E) Slitrk5, and F) Slitrk6. The intricate organization and regulation of synaptic connections involve complex and coordinated molecular and cellular processes. Diverse protein-protein interactions and their functional impacts at synapses. To explore the functions of Slitrks and their interactions with other proteins, we utilized the online String tool to predict protein-protein interactions among closely related proteins. Slitrk1, Slitrk2, and Slitrk3 were closely associated with PTPRD, PTPRS, PTPRF, NLGN1, NTR3, and IL1RAL1. Notably, Slitrk3 also exhibited close interactions with NTRK1, NRXN1, and NRXN2; Slitrk4 showed close interactions with PTPRD, NTRK1, ADGRL3, and EPHA6; Slitrk5 demonstrated close interactions with PTPRD, PTPRS, NTRK1, NTRK2, and RAC3; and Slitrk6 interacted closely with PTPRD and BDNF.

Figure 3. Graphical representation of the molecular organization of excitatory and inhibitory synapses. The regulation of neural circuit functions and behaviors is dictated by numerous synaptic organizers that maintain the equilibrium between excitatory and inhibitory synaptic inputs. Among Slitrk family proteins certain Slitrk isoforms- Slitrk1, 2, 4, and 5 were found to selectively function at excitatory synapses in diverse functional assays, whereas Slitrk3 specifically operated at inhibitory synapses. As shown in the figure, Slitrk1, 2, 4, and 5 present excitatory synapses; however, Slitrk 3 is an inhibitory synapse as an adhesion molecule (created with Biorendor.com).

Figure 4. BDNF signaling & Slitrk5 (adopted by [49] and created by Biorendor.com). Slitrk5, a cell-surface protein with LRR motifs, acts as a co-receptor that enhances TrkB signaling. TrkB and Slitrk5 physically interact by forming a complex involving their LRR domains. TrkB and PTPδ compete for binding to the initial LRR domain of Slitrk5. Slitrk5 influences BDNF-mediated biological reactions by directly controlling the recycling of the TrkB receptor through the recruitment of Rab11-FIP3.

We hope the above responses have clarified our approach and addressed all your concerns. We are grateful for your constructive feedback and appreciate your consideration of our manuscript.

Reviewer 3 Report

Comments and Suggestions for Authors

The article by Nidhi Puranik and Minsok Song is a literature review describing the role of Slitrk proteins, a family of transmembrane proteins with leucine-rich repeats, in CNS development. Slitrk protein variations have been associated with various sensory and neuropsychiatric disorders. The role of Slitrk family proteins in CNS development is diverse and still remains poorly understood.

The authors also described the features of the Slitrk family proteins. Slitrk1, 2, 4 and 5 are present in excitatory synapses (glitamate), Slitrk - in inhibitory synapses (glycine and GABA). Table 5 summarizing the possible contribution of mutations in Slitrk family genes to the development of CNS-associated diseases is extremely interesting and useful. The manuscript systematizes studies on the role of the Slitrk family of proteins in the development of CNS diseases, not only in patients but also in various animal models including zebrafish and pigs.

The review provides a comprehensive understanding of the structure and function of the investigated protein family as well as their possible role in disease mechanisms. The authors also highlight gaps in the existing knowledge and state the need for further research on this topic.

The manuscript may be published with minimal revisions. The only remark concerns citations (reference 11).

P.2, line 69. “These three disorders are all caused by the dysfunction of a neurotransmitter called serotonin [11]”. Reference [11] focuses on the combination of Western and Chinese medicine in the treatment of a number of neuropsychiatric disorders, including obsessive-compulsive disorder, anxiety disorder and depression. However, it is incorrect to refer to it as an expert opinion on the mechanisms underlying these disorders.

Author Response

Response: Thank you so much for providing your valuable time in reviewing the manuscript. We tried to incorporate all your comments into the revised version of the manuscript.

The article by Nidhi Puranik and Minsok Song is a literature review describing the role of Slitrk proteins, a family of transmembrane proteins with leucine-rich repeats, in CNS development. Slitrk protein variations have been associated with various sensory and neuropsychiatric disorders. The role of Slitrk family proteins in CNS development is diverse and still remains poorly understood.

The authors also described the features of the Slitrk family proteins. Slitrk1, 2, 4 and 5 are present in excitatory synapses (glitamate), Slitrk - in inhibitory synapses (glycine and GABA). Table 5 summarizing the possible contribution of mutations in Slitrk family genes to the development of CNS-associated diseases is extremely interesting and useful. The manuscript systematizes studies on the role of the Slitrk family of proteins in the development of CNS diseases, not only in patients but also in various animal models including zebrafish and pigs.

The review provides a comprehensive understanding of the structure and function of the investigated protein family as well as their possible role in disease mechanisms. The authors also highlight gaps in the existing knowledge and state the need for further research on this topic.

Response: Thank you for your review of our article. We appreciate your acknowledgment of the focus on Slitrk proteins and their role in CNS development. As highlighted, while the function of Slitrk proteins in this context is indeed diverse and not yet fully understood, we hope that our literature review provides valuable insights into their potential roles and associations with various sensory and neuropsychiatric disorders.

The manuscript may be published with minimal revisions. The only remark concerns citations (reference 11).

P.2, line 69. “These three disorders are all caused by the dysfunction of a neurotransmitter called serotonin [11]”. Reference [11] focuses on the combination of Western and Chinese medicine in the treatment of a number of neuropsychiatric disorders, including obsessive-compulsive disorder, anxiety disorder and depression. However, it is incorrect to refer to it as an expert opinion on the mechanisms underlying these disorders.

Response: We completely agree with the reviewer's suggestion and thank you for highlighting this major issue. The statement for reference 11 has been modified accordingly.

‘These three disorders are caused by the changes or damage of part of the brain and nervous system (11)’.

If you have any further questions or suggestions for enhancing the manuscript, we would be happy to address them.

Round 2

Reviewer 2 Report

Comments and Suggestions for Authors

The authors considered all reviewer comments. The current version of the manuscript has been improved, and all the reviewers' concerns have been addressed. I have no additional comments on the revised version of the manuscript.

Author Response

Thank you for your thoughtful review and feedback. I’m glad to hear that the revised manuscript meets the reviewers’ expectations and that all concerns have been effectively addressed.